# From Childhood Obesity to Metabolic Dysfunction-Associated Steatotic Liver Disease (MASLD) and Hyperlipidemia Through Oxidative Stress During Childhood

**DOI:** 10.3390/metabo15050287

**Published:** 2025-04-24

**Authors:** Siham Accacha, Julia Barillas-Cerritos, Ankita Srivastava, Frances Ross, Wendy Drewes, Shelly Gulkarov, Joshua De Leon, Allison B. Reiss

**Affiliations:** 1Department of Pediatrics, NYU Grossman Long Island School of Medicine, Mineola, NY 11501, USA; siham.accacha@nyulangone.org (S.A.); julia.barillascerritos@nyulangone.org (J.B.-C.); frances.ross@nyulangone.org (F.R.); 2Department of Foundations of Medicine, NYU Grossman Long Island School of Medicine, Mineola, NY 11501, USA; ankita.srivastava@nyulangone.org (A.S.); shellygulkarov@mail.adelphi.edu (S.G.); 3Department of Medicine, NYU Grossman Long Island School of Medicine, Mineola, NY 11501, USA; wadrewes27@gmail.com (W.D.); joshua.deleon@nyulangone.org (J.D.L.)

**Keywords:** metabolic dysfunction-associated fatty liver disease, obesity, pediatric obesity, reactive oxygen species (ROS), inflammation

## Abstract

Background/Objectives: Metabolic dysfunction-associated steatotic liver disease (MASLD), previously known as non-alcoholic fatty liver disease (NAFLD), is rapidly becoming the most prevalent form of chronic liver disease in both pediatric and adult populations. It encompasses a wide spectrum of liver abnormalities, ranging from simple fat accumulation to severe conditions such as inflammation, fibrosis, cirrhosis, and liver cancer. Major risk factors for MASLD include obesity, insulin resistance, type 2 diabetes, and hypertriglyceridemia. Methods: This narrative review employed a comprehensive search of recent literature to identify the latest studies on the relationship between MAFLD and obesity, the health consequences and the latest treatment options to prevent long-term damage to the liver and other organs. Additionally, the article presents perspectives on diagnostic biomarkers. Results: Childhood obesity is linked to a multitude of comorbid conditions and remains a primary risk factor for adult obesity. This abnormal fat accumulation is known to have long-term detrimental effects into adulthood. Scientific evidence unequivocally demonstrates the role of obesity-related conditions, such as insulin resistance, dyslipidemia, and hyperglycemia, in the development and progression of MASLD. Oxidative stress, stemming from mitochondrial dysfunction, is a leading factor in MASLD. This review discusses the interconnections between oxidative stress, obesity, dyslipidemia, and MASLD. Conclusions: Atherogenic dyslipidemia, oxidative stress, inflammation, insulin resistance, endothelial dysfunction, and cytokines collectively contribute to the development of MASLD. Potential treatment targets for MASLD are focused on prevention and the use of drugs to address obesity and elevated blood lipid levels.

## 1. Introduction

Childhood obesity is a multifactorial nutrition-related disorder characterized by a disruption in regulating energy balance. It is one of the most important risk factors for developing various metabolic diseases, such as type 2 diabetes mellitus, cardiovascular diseases, and liver disease [1,2]. In fact, numerous studies suggest a strong association between obesity and insulin resistance, a condition characterized by decreased sensitivity to insulin in adipose tissue, liver, and skeletal muscle [3].

Obesity-induced insulin resistance is linked to a wide cluster of impairments, such as dyslipidemia, metabolic dysfunction-associated steatotic liver disease (MASLD), hypertension, coronary heart disease, stroke, and neurodegeneration [4]. The risk of developing MASLD is elevated in children who meet the criteria for obesity, and in persons with type 2 diabetes, inadequately controlled type 2 diabetes leads to more extensive liver disease compared with patients without diabetes [5,6,7,8]. Due to the rise in obesity, MASLD has become a major public health problem in adults and children in industrialized countries, where children have a sedentary lifestyle and consume a high amount of ultra-processed foods. Diagnosing and staging MASLD in clinical practice remains complex, but it is a diagnosis that should not be missed [9]. In addition to liver-related morbidity, MASLD is also associated with an increased risk of cardiovascular disease, type 2 diabetes, and mortality in adulthood. Therefore, pediatric patients with MASLD are of particular concern because, with a long lifespan ahead, more complications will have a chance to accrue. Moreover, various forms of metabolic disorders, including pre-diabetes, metabolic syndrome, and high-fat-diet-induced obesity, have been linked to MASLD [10,11,12].

Recent evidence indicates that organs outside the liver can affect the progression of MASLD [13,14,15,16]. Adipose tissue dysfunction and insulin resistance are significant events in the development of MASLD, highlighting the connection between adipose tissue and the liver [13,14].

The present review aims to provide a broad summary of the common factors relating to childhood obesity, oxidative stress, insulin resistance, and MASLD in the pediatric population. It focuses on how insulin resistance and oxidative stress may play a role in causing the alterations in the hepatic structure characteristic of MASLD. In addition, a comprehensive overview of the most up-to-date therapeutic approaches in the pediatric population is highlighted.

## 2. Obesity in Children—Scope of the Problem

### 2.1. Prevalence and Incidence

Pediatric obesity is a worldwide health problem with mounting prevalence in low-, middle-, and high-income countries [17,18]. In October 2019, the World Obesity Federation published estimates of childhood obesity prevalence from 2020 to 2030 [19]. They projected 206 million children and adolescents aged 5–19 years will be living with obesity in the year 2025 and 254 million by 2030. The United States (US) is ranked third on the list of countries predicted to have over 1 million school-aged children living with obesity by 2030.

In a systemic review and meta-analysis of 1668 studies, including over 44 million individuals, Zheng et al. found the prevalence of obesity in children under the age of 18 years to be 8.5% globally and 18.6% in the US [20]. When stratified by geographical regions, the highest prevalence of obesity was found in Polynesia, with an estimated rate of 19.5%, and the lowest was in Middle Africa at 2.4%. The study also found a positive association between a country’s income and the prevalence of obesity in children. High-income countries have the highest prevalence, while low-income countries show the lowest.

Studies agree that there are disparities in pediatric obesity prevalence based on specific demographic factors [21,22]. There is a greater prevalence in male than female children [18,23,24]. Notably, the Non-Communicable Diseases Risk Factor Collaboration reported approximately 5.6% of girls and 7.8% of boys in 2016 were living with obesity [25]. Additionally, several studies have revealed a lower prevalence of obesity in adolescents than in preschool and school-aged children [20,26]. This decline in obesity prevalence may be secondary to pubertal hormone shifts and the increased consciousness of teens about their appearance.

Skinner et al. examined the prevalence of obesity and severe obesity in US children from 1999 to 2016 using the National Health and Nutritional Examination Surveys (NHANES) [27]. Ultimately, one in five children and adolescents suffer from obesity in the US, a statistic that has more than tripled since the 1970s. Non-Hispanic African American and Hispanic children had the highest prevalence rates of obesity when compared to all other races [28]. Their study also highlighted a significant increase in obesity and severe obesity in children aged 2–5 and adolescent females aged 16 to 19 years, specifically in the years 2015 to 2016 [27]. Despite strong clinical efforts and an immense public health focus on obesity in the US, prevalence remains high and continues to grow at an alarming rate.

### 2.2. Etiology

Childhood obesity is a multifactorial disease with numerous biological, behavioral, and environmental causes [29,30]. Environmental influences such as family, community, and school have been shown to contribute to excess weight gain in children and adolescents in various ways [31,32,33]. The eating behaviors formed in childhood are strongly associated with parental psychosocial state and feeding styles. Excessive stress and depression in parents can lead to maladaptive coping strategies such as eating to manage these emotions [34].

Other eating patterns associated with obesity include skipping breakfast, consuming excess fat and refined carbs, and unnecessary intake of sugar drinks [35,36,37]. Trends in the US show that American children have diets higher in processed foods that are known to be filled with added sugars and unhealthy fats [38,39,40]. It is estimated that 10 to 15% of the total caloric intake of children in the US comes from sugar-sweetened beverages, averaging 270 kcal/d. [41], and a population-based study of young persons aged 3–19 years encompassing data from 185 countries showed a 23% increase in consumption of beverages with added sugar over the years 1990 to 2018 [42].

Another etiological factor contributing to childhood obesity is increased sedentary lifestyle [43]. Children and adolescents spend more time watching television and using computers, phones, and tablets [44].

Reduced physical activity intensified during the COVID-19 pandemic and showed a rapid increase in the onset of pediatric obesity [45,46]. Multiple factors contributed to the increase in obesity prevalence during the COVID-19 lockdown period. Chang et al. wrote a systemic review and performed a meta-analysis in 2021 to quantify the effects of this time on the body weight of pediatric patients. They found an overall increase in body weight and body mass index (BMI) in school-aged children [46]. A regional study in the Children’s Hospital of Philadelphia Care Network encompassing urban, suburban, and semi-rural areas showed that obesity prevalence in children increased from 13.7% to 15.4% in the timespan from January 2019 to December 2020 [47]. Apart from decreased physical activity, worsening dietary habits fueled by boredom and stress occurred during the pandemic. A multinational examination showed an increased intake of processed meat, snacks, sugary beverages, and potato chips [48]. COVID-19 lockdown had a significant impact on the lifestyles of children that worsened the pre-existing obesity pandemic [49].

Inherited genetic syndromes and single gene defects make up a small percentage of children with obesity [50]. Prader–Willi syndrome is one of the most recognized and common syndromes associated with childhood obesity [51]. Children will often present with hypotonia and feeding difficulties in infancy, followed by a phase of hyperphagia in early childhood. During this phase, children and adolescents develop obesity [52]. Monogenic obesity has a severe, early onset and is typically due to an autosomal recessive mutation in the leptin-melanocortin pathway genes leading to increased food-seeking behaviors [53].

Other secondary causes of childhood obesity can be endocrine, neurologic, medication-induced, or hypothalamic in origin (Table 1) [54]. These are typically due to a hormonal imbalance that leads to unexpected weight gain [55].

### 2.3. Obesity, Metabolic Syndrome, and Chronic Systemic Inflammation

As the prevalence of pediatric obesity rises, the relationship with metabolic syndrome becomes clearer [56]. While specific endpoint events can be evaluated in adults, pediatric studies focus on known adult cardiometabolic risk factors.

Metabolic syndrome in childhood, also known as “Insulin Resistance Syndrome”, encompasses multiple factors that promote the development of atherosclerotic cardiovascular disease and type 2 diabetes in adulthood [57,58]. These factors include low HDL, elevated triglycerides, elevated glycated hemoglobin levels, visceral adiposity, and elevated blood pressure [59].

Insulin resistance induced by inflammatory mediators is a key factor in the development of metabolic dysfunction-associated steatotic liver disease (MASLD) [60]. Insulin binds to cell receptors, causing the receptor to phosphorylate itself and other insulin receptor substrates (IRSs). In the liver, the main mediators of insulin signaling are insulin receptor substrate 1 (IRS1) and insulin receptor substrate 2 (IRS2) [61]. Insulin resistance is defined as the impairment of IRS1- or IRS2-dependent signaling pathways. These pathways include the use of phosphatidylinositol 3-kinase (PI3K)–phosphoinositide-dependent kinase (PDK)–protein kinase B (AKT) and the rat sarcoma (RAS)−extracellular signal-regulated kinase (ERK). The PI3K-PDK-AKT pathway mediates glycogen synthesis and gluconeogenesis, and the RAS-ERK pathway mediates cell proliferation and survival. When the downstream effects of insulin signaling in the muscle, liver, and adipose tissue are significantly impaired, lipid deposition, immune-mediated inflammatory changes, and diminished disposal of glucose lead to MASLD [62].

Insulin resistance is believed to start in the muscle tissue and then adipose tissue [63]. In adipose tissue, insulin resistance contributes to lipolysis in adipocytes and increased free fatty acid circulation, ultimately worsening liver steatosis. When obese patients with insulin resistance take in calories, glycogenolysis is not appropriately inhibited. Despite caloric intake and postprandial glucose rising in the blood, hepatic glucose production continues, leading to glucotoxicity [64]. This turns into a repetitive cycle and re-contributes to insulin resistance.

While insulin resistance is a considerable part of metabolic syndrome, chronic low-grade systemic inflammation in obese patients is a major driving force for the acceleration of atherosclerotic disease and risk of mortality in adulthood [65]. Many studies have defined elevated levels of cytokine and acute phase reactants due to excess adipose tissue [66]. The tissue secretes adipokines that alter the regulation of insulin sensitivity, glucose and lipid metabolism, endothelial function, and overall inflammation.

Adipose tissue contains various immune cells. As adipose tissue accumulates, an increase in cytokines such as tumor necrosis factor-α (TNF-α) and interleukin (IL)-6, as well as leptin, resistin, and lipocalin 2, has been shown to play an important role in insulin resistance and inflammation [67]. It was noted to be linked to infiltration and accumulation of macrophages within the tissue, more specifically, classically activated macrophages (M1-macrophages) or those that express markers F4/80, a microglial plasma membrane glycoprotein, and CD11c, a transmembrane glycoprotein found on monocytes and macrophages. Visceral adipose tissue, characterized as having a higher number of larger adipocyte cells and lower insulin sensitivity, was noted to have a stronger impact on overall metabolism [68,69]. This may be due to a predominantly M1-macrophage infiltration that is known to primarily express TNF-α and inducible nitric oxide synthase that directly affects the insulin sensitivity of the cells [70]. TNF-α and IL-6 stimulate phosphorylation of serine residues instead of tyrosine on IRS-1. This phenomenon leads to the inhibition of insulin signaling [Figure 1].

Adipose tissue hypertrophy and subsequent dysfunction also contribute to the development of MASLD. Leptin and adiponectin in adipose tissue cause uninhibited lipolysis [71]. This occurrence, along with excessive dietary intake, leads to increased levels of free fatty acids in the bloodstream [72]. The liver’s response is to increase the uptake of free fatty acids from the circulation and continue de novo hepatic lipogenesis. As free fatty acids accumulate in the liver, they oxidize and become stored as triglycerides and very low-density lipoproteins. When hepatocyte metabolic capacity is exceeded, mitochondrial dysfunction ensues, leading to the overproduction of reactive oxygen species (ROS) [73]. The endoplasmic reticulum within hepatocytes is also damaged by ROS, causing an accumulation of improperly folded, inactive proteins. Lipotoxicity results in hepatocyte inflammation, apoptosis, and eventual fibrosis in the livers of pediatric patients with obesity [74].

In pediatrics, the question remains: when does chronic inflammation begin and start to take effect? With the increased prevalence of obesity across all ages, inflammatory changes may occur in the earliest developmental stages. For example, neonates with low birth weight and intrauterine growth restriction have increased inflammatory markers, creating a strong risk factor for future metabolic disease [75]. Children as young as 3 years old demonstrate elevated c-reactive protein (CRP) and absolute neutrophil counts that correlate with many metabolic syndrome mechanisms [70]. Complications arising from chronic low-grade systemic inflammation in pediatric patients have tangible consequences that many studies predict will lead to shorter life expectancy in adulthood [68].

### 2.4. Hormones and Obesity

Although the explanation that excess energy intake or decreased energy expenditure leads to weight gain is attractive in its simplicity, research during the last decade shows that appetite regulation and energy homeostasis rely on multiple hormones, organs, and neural tracts, which together encompass the appetite regulatory system [76]. The hypothalamus and the nucleus of the solitary tract (NTS) primarily regulate appetite and energy intake. The NTS, located within the hypothalamus, is a crucial nucleus that receives and integrates signals from peripheral organs, particularly the digestive tract and the central nervous system, to participate in appetite regulation. Furthermore, the NTS interacts with other hypothalamic nuclei, particularly the arcuate nucleus, to further regulate appetite. Within the arcuate nucleus, two distinct types of neurons can be found: orexigenic (promoting appetite) and anorexigenic (suppressing appetite). Orexigenic neurons express appetite-stimulating signal substances such as neuropeptide Y and Agouti-Related Protein (AgRP). Conversely, anorexigenic neurons express appetite-suppressing signal substances like proopiomelanocortin and cocaine and amphetamine-regulated transcript peptide (CART). Additionally, the dopaminergic reward system in the midbrain, which controls feeding reward regulation, can continuously promote appetite by providing a sense of satisfaction and pleasure. This system regulates appetite through the acquisition of reward and hedonic value [77,78]. The gastrointestinal tract and adipose tissue emit appetite regulation, food sensing, and digestion signals in the peripheral system. Ghrelin is an orexigenic or appetite-stimulating gut hormone, and it is secreted primarily by the oxyntic glands of the stomach [77]. Meanwhile, appetite inhibition and satiety are signaled in the gastrointestinal tract by the action of cholecystokinin (CCK), glucagon-like peptide 1 (GLP-1), and peptide YY. Adipose tissue participates in long-term weight regulation by secreting leptin, a hormone that regulates long-term appetite [78].

### 2.5. Associated Comorbidities of Childhood Obesity

Pediatric obesity can have lasting effects on all body systems. In 2019, a review of 52 studies involving over 1.5 million participants reported that children and adolescents with obesity had a higher prevalence of diabetes, hypertension, dyslipidemia, depression, and MASLD than their healthy-weight peers [4,20,23].

Cardiometabolic and cardiovascular comorbidities have been well-established in children diagnosed with obesity [79]. Obese children were found to have left ventricular and atrial hypertrophy and systolic and diastolic dysfunction on echocardiogram [80,81]. This was shown in a retrospective cross-sectional analysis of anthropometric and M-mode echocardiography data performed in healthy children and adolescents separated by one generation. Children from 1986 to 1989 were compared to those from 2008 in over 11,000 participants. One defining difference between the two groups was the elevated BMI in the more current population. Elevated childhood adiposity and BMI predict cardiac mass in adulthood; thus, an increase in weight can predict which children will develop left ventricular hypertrophy and be at increased risk of stroke, sudden death, myocardial infarction, coronary artery disease, and congestive heart failure. Hypertension is much more prevalent in obese children. It may contribute to the development of excess left ventricular mass [82].

Other cardiometabolic comorbidities stem from hyperinsulinemia and the subsequent development of prediabetes and type 2 diabetes mellitus [83]. Adolescents who develop type 2 diabetes are known to have a more rapid decline in glycemic control and progression of complications. Diabetes-related complications include microalbuminuria, hypertension, and dyslipidemia [84].

The endocrine system is strongly affected by pediatric obesity. Outside of insulin resistance, studies have also shown the role of obesity in the pathogenesis of polycystic ovarian syndrome [85]. Hyperinsulinemia activates ovarian androgen production and could potentially be a driving factor in the development of polycystic ovarian syndrome, especially in females with genetic predisposition [86].

Children with obesity have multiple pulmonary comorbidities. There is a higher prevalence of obstructive sleep apnea, leading to chronic fatigue and poor concentration [87,88]. Childhood obesity has also been shown to be strongly associated with asthma and worsening uncontrolled asthma [89].

Other well-known comorbidities include musculoskeletal, psychosocial, dermatologic, and neurologic. Childhood obesity increases the risks of fractures, slipped capital femoral epiphysis, and joint pain [90,91]. A 2018 cohort study showed that children aged 11 to 12 years old with severe obesity have a lifetime risk of developing a slipped capital femoral epiphysis that is 17.0 times greater than those with normal BMI. Several studies have revealed a possible mechanism in children with obesity. Obese children generate forces sufficient to overcome the yield point of the physis. These forces in combination with rapid growth during puberty are a plausible mechanism for obesity causing a slipped capital femoral epiphysis [92]. The inflammatory state conferred by obesity may be detrimental to bone quality, leading to fractures despite the stimulating effect of weight-bearing on bone production [93,94]. Children who suffer from obesity and overweight commonly suffer from anxiety, depression, and low self-esteem. Many children experience bullying and discrimination in school [95]. Obese children often develop acanthosis nigricans suggestive of insulin resistance and are at increased risk for intertrigo and hidradenitis suppurativa [96]. These dermatological abnormalities contribute to psychosocial comorbidities. Obesity in children and adolescents is associated with a higher incidence of pseudotumor cerebri [97]. Children will present with headaches, vomiting, and vision changes, sometimes requiring diagnostic and therapeutic lumbar puncture.

In pediatric patients with obesity, the prevalence of MASLD is estimated to exceed 20% [98]. Patients typically present as asymptomatic, but liver enzyme levels on lab work will be elevated. MASLD increases the overall risk of cirrhosis and hepatocellular carcinoma in adulthood. In adults, MASLD is known to contribute to the development of chronic kidney disease, cardiovascular disease, and type 2 diabetes. Risk factors for the development of MASLD in children with obesity include male sex, the start of puberty, Hispanic ethnicity, intestinal dysbiosis, positive family medical history, and exposure to endocrine-disruptive chemicals in the environment. Genetic variants in patatin-like phospholipase domain-containing protein 3 (PNPLA3), transmembrane 6 superfamily member 2 (TM6SF2), glucokinase regulatory protein (GCKR), and membrane-bound o-acyltransferase 7 (MBOAT7) also increase the risk of MASLD [99].

Obesity is a worldwide pandemic with biological, behavioral, and environmental causes. Children living with obesity suffer multi-system comorbidities both in the short and long term.

## 3. Obesity and MASLD in Children—Clinical Diagnosis and Metabolic Consideration

### 3.1. Definition of Metabolic Dysfunction-Associated Steatotic Liver Disease MASLD

Before delving into the complexities of steatotic liver disease, it is essential to clarify its evolving nomenclature. In the early 1980s, Ludwig et al. introduced the term non-alcoholic fatty liver disease (NAFLD) to describe hepatic steatosis occurring in the absence of significant alcohol consumption [100,101]. However, as the understanding of the disease expanded, it became evident that this terminology did not fully capture its metabolic implications or clinical heterogeneity. Additionally, concerns regarding its exclusionary and potentially stigmatizing nature prompted efforts to refine its classification.

In 2023, a global Delphi consensus process, led by three major liver associations, resulted in the reclassification of NAFLD as metabolic dysfunction-associated steatotic liver disease (MASLD) to better reflect its association with cardiometabolic risk factors. Previously, NAFLD was categorized into non-alcoholic fatty liver (NAFL) and non-alcoholic steatohepatitis (NASH) based on histological features. NAFL, defined by the presence of at least 5% hepatic steatosis without hepatocellular injury (e.g., hepatocyte ballooning), has been renamed MASL (metabolic dysfunction-associated steatotic liver), representing a non-progressive subtype of MASLD. Conversely, NASH—characterized by hepatic steatosis with inflammation and hepatocyte injury, with or without fibrosis—has been redefined as metabolic dysfunction-associated steatohepatitis (MASH), signifying the progressive form of MASLD [101,102].

Additionally, a new classification, MetALD (metabolic dysfunction-associated liver disease with significant alcohol intake), has been introduced to account for individuals with MASLD who also consume alcohol at levels that exceed previously established thresholds. This refined terminology provides a more precise and inclusive framework for diagnosing and managing steatotic liver disease. MASLD is now recognized as the most prevalent chronic liver disease in industrialized nations, closely linked to the obesity epidemic [102,103].

In children, diagnosing MASLD has proven challenging because there is no definitive non-invasive diagnostic tool, and effective treatment options are lacking [104]. The 2017 NASPGHAN clinical practice guideline recommends the most comprehensive screening approach for children (8–18 years) with obesity or overweight plus an additional risk factor (family history of MASLD, insulin resistance, (pre)diabetes, dyslipidemia, or central adiposity). ALT is the primary screening tool for steatosis, with further evaluation advised if ALT remains elevated (≥2× ULN: boys ≥ 52, girls ≥ 44 IU/L) after 3–6 months of lifestyle intervention or immediately if ALT ≥ 80 IU/L. For normal results, NASPGHAN and most guidelines recommend repeating screening every 2–3 years [105].

Diagnostic criteria for MASLD include evidence of intrahepatic fat (via biopsy, imaging, or validated blood biomarkers) plus one of the following: excess adiposity, prediabetes or type 2 diabetes, or metabolic dysregulation (defined as at least two metabolic risks, such as increased waist circumference, hypertension, abnormal lipid profiles, or impaired fasting glucose) [102,106]. Ethnicity-specific cutoffs may apply [107]. In the adult population, another distinction is lean MASLD, which is less evident in children. Schwimmer et al. found that 5% of normal-weight children exhibited histological evidence of MASLD in an autopsy study conducted in San Diego [108,109]. This highlights the lack of correlation between BMI and MASLD, as BMI does not accurately reflect liver steatosis [110]. Distinct patterns of liver histology, such as “portal predominant” inflammation, are more common in children than adults. Still, the diagnosis of MASLD is one of exclusion as a systematic evaluation is required to rule out secondary causes such as metabolic disorders or viral hepatitis. Imaging and histopathological evaluation play crucial roles, with liver biopsy reserved for cases where diagnosis remains unclear [102,111]. Steatosis is defined as abnormal fat accumulation within hepatocytes, and the minimum criterion for MASLD diagnosis is the presence of macrovesicular steatosis (fat droplet equal to or larger than the size of the nucleus, often displacing the nucleus) or microvesicular steatosis (small lipid vesicles in the cytoplasm) in >5% of hepatocytes [106,112].

In pediatric populations, MAFLD shows a clear sex-related disparity, with males consistently exhibiting higher prevalence than females. Studies report rates of 24.5–35.3% in males versus 12.0–24.4% in females, particularly among children with obesity. This difference is likely driven by variations in fat distribution, sex hormones, and metabolic factors, underscoring the need to consider sex as a key variable in both assessment and management, particularly when employing a tailored, precision-medicine approach to treatment [111].

Standardized reference measures for children are available for blood pressure, plasma triglycerides, HDL-cholesterol, fasting glucose, 2 h post-glucose load, hemoglobin A1c, and CRP. Age- and sex-specific reference values are recommended to ensure accurate assessment. Waist circumference, which demonstrates a stronger correlation with central obesity than BMI, should also be incorporated into evaluations [113,114].

Although uric acid is not routinely measured in pediatric populations, it provides valuable insight into the pathophysiology of MASLD. Uric acid, a byproduct of sucrose metabolism via the keto hexokinase (KHK) pathway which involves purine degradation and uric acid synthesis during fructolysis, indicates increased fructose metabolism [115]. Greater fructose metabolism contributes to inflammation, oxidative stress, and hepatic fat deposition [114]. Elevated uric acid levels are associated with MASLD, even though they are not commonly tested. Alanine aminotransferase (ALT), a sensitive marker of hepatocyte injury, when elevated more than twice the upper limit of normal (<26 U/L for boys, <22 U/L for girls), strongly suggests liver inflammation and potential progression to steatohepatitis [115,116]. Liver biopsy remains the gold standard for diagnosing MASLD but is rarely performed due to its invasiveness and risks of sampling or interpretation errors. North American Society of Pediatric Gastroenterology, Hepatology and Nutrition (NASPGHAN) guidelines recommend liver biopsy for patients at high risk of MASH and advanced fibrosis, such as those with ALT levels greater than 80 U/L, splenomegaly, or an aspartate aminotransferase (AST)-to-ALT ratio above 1 [104,117]. Liver biopsy should be considered in lean children with features that may indicate an alternative diagnosis.

### 3.2. Genesis of MASLD

Pediatric metabolic-associated steatotic liver disease (MASLD) is becoming a significant consequence of the global increase in childhood obesity [111]. This rise is driven by a complex interaction of genetic, metabolic, and environmental factors. Understanding the early origins of MASLD—from infancy through episodes of rapid weight gain in childhood—is essential for developing effective prevention and intervention strategies. The Avon Longitudinal Study of Parents and Children (ALSPAC) study revealed that increased weight gain relative to height (adiposity gain) during childhood, particularly from the age of 3 onward, and a higher BMI at ages 7 and 10 are strongly linked to liver damage markers in adolescence [118,119]. These findings underscore the importance of early interventions to prevent excessive weight gain in childhood, thereby reducing the risks of developing steatotic liver disease in adolescence and possibly into adulthood. While previous research has suggested that infancy may be a critical period for weight-related cardiometabolic risks, this study found no consistent connection between infant BMI and liver outcomes. Instead, steady increases in adiposity between ages 1 and 10 were consistently associated with a higher risk of MASLD in adolescence, with the risk increasing over time. These findings highlight the importance of early and sustained efforts to manage weight gain in young children in order to prevent excessive body fat accumulation and future liver disease [118]. Similarly, a diet rich in fructose, foods with a high glycemic index, and sugar-sweetened drinks have a significant role in the development of a steatotic liver, which highlights the association between diet and MASLD [120]. The fructose component of sugar and high fructose corn syrup induces fatty liver by stimulating de novo lipogenesis and blocking β-fatty acid oxidation [121]. These effects have been attributed to fructose metabolism by fructokinase, which leads to a fall in adenosine triphosphate (ATP) with nucleotide turnover and uric acid generation. This, in turn, has prooxidative and pro-inflammatory effects that exacerbate the lipogenic process in the liver [122].

Emerging evidence suggests that individuals with celiac disease, a systemic autoimmune disorder triggered by consumption of gluten, have a higher incidence of hepatic steatosis when adhering to a gluten-free diet compared to the general population. A prospective case–control study examining 202 patients with celiac disease on a gluten-free diet and 202 matched controls found that celiac disease patients had a threefold increased risk of developing a steatotic liver, despite being matched in a 1:1 ratio based on age, gender, and traditional metabolic risk factors such as overweight, diabetes mellitus, total cholesterol, and triglycerides. Notably, this increased risk was also observed in lean individuals and those with fewer metabolic risk factors, suggesting an independent link between celiac disease and hepatic steatosis [123]. One proposed mechanism underlying this association is the disruption of the gut–liver axis in celiac disease, leading to increased intestinal permeability and small intestinal bacterial overgrowth—both of which have been implicated in metabolic-associated liver steatosis. Small intestinal bacterial overgrowth is a recognized contributor to hepatic steatosis and is characterized by an imbalance in the gut microbiome that weakens the mucosal barrier, heightening its permeability. This process facilitates the translocation of pathogenic microbiota and their metabolites into the systemic circulation, potentially exacerbating liver inflammation and metabolic dysfunction [124]. Although MASLD is often associated with obesity and metabolic syndrome, the increased prevalence of hepatic steatosis in celiac disease patients suggests that alternative pathways beyond traditional metabolic factors may be involved. Mechanisms that have been postulated to contribute include disruption of the gut microbiome and changes in intestinal permeability in those with celiac disease [125]. This underscores the need for further research into gut dysbiosis and its role in the pathogenesis of hepatic steatosis in celiac disease, particularly in individuals without conventional metabolic risk factors.

An animal study revealed that increased intestinal gluconeogenesis helps to regulate glucose levels and prevents the development of hyperglycemia, even when following a high-fat and high-sucrose diet. It mainly affects the onset of hepatic steatosis and the molecular processes associated with the development of MASLD through a neural circuit that connects the gut, brain, and liver [15]. On the other hand, suppressing intestinal gluconeogenesis leads to increased lipid accumulation in the liver, even when following a standard diet. Intestinal gluconeogenesis offers metabolic benefits by initiating a gut–brain neural signal, which triggers brain-dependent regulation of peripheral metabolism [15,16].

Most recently, Goyal et al. genotyped 822 children with biopsy-confirmed MASLD with allele-specific primers for 60 candidate single-nucleotide polymorphisms and enrolled parents from trio analysis. PNPLA3 (rs738409) demonstrated the strongest risk for NAFLD, and PARVB (rs6006473) was highly associated with the severity of fibrosis [126].

In summary, the shift from NAFLD to MASLD reflects a deeper understanding of liver disease, emphasizing its connection to metabolic syndrome. MASLD incorporates a broader range of metabolic factors and diagnostic criteria, moving beyond simple fat accumulation. This redefinition reduces stigma and better reflects the disease’s systemic nature. As research progresses, it highlights the need to address the multifactorial causes of MASLD.

## 4. Oxidative Stress-Induced Pathologic Changes in MASLD

### 4.1. Oxidative Stress in Childhood MASLD

Oxidative stress or redox imbalance is an incongruence between cellular antioxidants and pro-oxidants (ROS and nitrogen species) that results in cellular damage and can ultimately lead to cell death. In normal physiological conditions, the balanced redox signaling in the cell is promoted by basal levels of cellular ROS which are required for various processes such as cell survival and differentiation, regulation of transcription factor activity, epigenetic changes, and cell metabolism [127]. ROS are mainly produced by mitochondria, peroxisomes, and the endoplasmic reticulum. Overproduction of ROS in these organelles alters their function and generates stress. The HO- (hydroxyl radical) is the dominant free radical species involved in the negative effects of oxidative stress. It induces lipid peroxidation and DNA breaks and damage [128]. Oxidative stress promotes the activation of robust antioxidant defensive pathways that neutralize ROS and preserve cellular integrity [129].

Experimental, clinical, and epidemiological studies have shown that MASLD is associated with high oxidative stress in liver cells, which leads to further metabolic risks and disease progression [130,131]. Kumar et al. found that the activity of enzymes involved in antioxidant mechanisms such as superoxide dismutase and glutathione peroxidase are decreased in patients with MASLD with a few exceptions in which activity of these enzymes was elevated [132]. A recent cross-sectional study in children found an inverse correlation between serum glutathione peroxidase and NAFLD [133]. Oxidative stress is highly prevalent in childhood MASLD [134]. Pirgon et al. showed increased oxidative stress index and total oxidant status in obese adolescents with MASLD [130].

Free fatty acids and their metabolites contribute to liver injury by increasing oxidative stress in children and adults with MASLD [135,136,137]. Several studies have shown that nutrient supplementation reduces oxidative stress, thus improving MASLD [134]. A variety of clinical trials have looked at the effect of specific supplements and found three regimens to be useful in reducing oxidative stress parameters in children with MASLD: (1) supplemental vitamin E plus hydroxytyrosol, (2) lycopene-rich tomato juice, and (3) omega-3 polyunsaturated fatty acids [138,139,140].

### 4.2. Liver Cells Involved in Oxidative Stress

#### 4.2.1. Hepatocytes

Hepatocytes are the primary cells of the liver parenchyma and constitute up to 80% of total liver cells [141]. Hepatocytes perform many vital functions such as detoxifying toxic metabolites and xenobiotic compounds, as well as metabolizing carbohydrates, proteins, and lipids. They are involved in the synthesis of serum proteins, lipoproteins, and complement components [142,143,144]. Hepatocytes help activate the innate immune response [145]. Homeostasis within hepatocytes is maintained through mitochondrial β-oxidation, electron transfer chain activity, ATP synthesis, ketogenesis, and the tricarboxylic acid cycle [146,147]. Hepatocytes are the primary site of ROS production, predominantly within the mitochondria, and thus, they are a major source of ROS-mediated oxidative stress [144]. Oxidative stress generated in the diseased liver in turn disrupts mitochondrial and endoplasmic reticulum functioning and affects fatty acid oxidation and lipid and protein synthesis [148,149,150]. Increased fatty acid levels in MASLD are responsible for the induction of fatty acid β-oxidation and accumulation of ROS as a byproduct of fatty acid metabolism. Excessive production and accumulation of ROS results in disordered lipid metabolism, inflammation, dysregulation in insulin sensitivity, and hepatocyte apoptosis [151,152,153]. High levels of ROS can accumulate during the breakdown of fatty acids into acetyl-CoA through β-oxidation, impeding mitochondrial function and further promoting the development of MASLD [154,155,156]. Mitophagy, the process of clearing mitochondria damaged by ROS, plays a protective role against oxidative damage in hepatic steatosis by alleviating fat accumulation, ROS generation, and inflammation [157,158]. Activation of hepatocyte apoptosis because of oxidative stress leads to the infiltration of Kupffer cells and activation of hepatic stellate cells (HSCs). Depletion of mitochondrial DNA, protein, and phospholipids due to overproduction of ROS results in mitochondrial dysfunction, which leads to the progression of hepatocytes apoptosis [159,160]. ROS excess results in the functional loss of mitochondria and other intracellular organelles such as peroxisomes and ER, which further perpetuates the activation of the apoptotic cascade and cell death [161,162].

#### 4.2.2. Kupffer Cells

Kupffer cells are the resident macrophage population residing in the liver. They play an important role in the innate immune response of the liver. Accumulation of fat during MASLD induces M1 polarization of Kupffer cells which elicits the production of inflammatory cytokines and ROS. Secreted pro-inflammatory cytokines activate HSCs and promote progression of hepatic steatosis to fibrosis [163,164]. ROS can damage Kupffer cell mitochondrial DNA, activating the innate immune response and initiating an inflammatory cascade of cell destruction [165]. Free fatty acids, cholesterol, and ROS contribute to the formation of foamy Kupffer cells. Kupffer cells sense free fatty acids and promote fat accumulation in hepatocytes via secretion of TNFα [166].

#### 4.2.3. Hepatic Stellate Cells

HSCs contribute to the minor population (5–10%) of liver cells. HSC activation leads to the development of hepatic fibrosis by promoting excess collagen formation and extracellular matrix accumulation. ROS-mediated oxidative stress plays a key role in activation of fibrogenesis in HSCs. Damaged hepatocytes and inflammatory KC release excess TGF-β which increases fibrogenesis in HSCs via activation of the SMAD2/3 pathway [167]. Intracellular increase in ROS levels is a critical event for directional migration of HSCs via activation of the ERK1/2 and JNK1/2 pathways [168]. Zhou et al. reported that oxidative stress induces mitochondrial fission, which further leads to activation of HSCs [169].

### 4.3. Inflammatory Pathways Associated with Oxidative Stress

Inflammation, advanced fibrosis, and cirrhosis are the progressive steps involved in transition from simple steatosis to MASH [170,171]. Kupffer cells and monocyte-derived macrophages are the distinct macrophage populations found in diseased liver and are responsible for liver inflammation and activation of HSCs during oxidative stress-induced MASH [172,173]. The activation of Kupffer cells marks the onset of hepatic inflammation in liver disease. Excess free fatty acids and cholesterol contribute to the formation of activated foamy Kupffer cells, which promote the progression of MASLD to steatohepatitis [174]. Free fatty acids are also responsible for generating endoplasmic reticulum stress and lipotoxicity in hepatocytes, which is further involved in the production of pro-inflammatory cytokines and the generation of extracellular vesicles that stimulate Kupffer cells [175,176]. Several studies have demonstrated that lymphocyte-derived cytokines such as IFN-γ, IL-17, and TNF superfamily member 14 (TNFSF14) can stimulate pro-inflammatory responses of monocyte-derived macrophages [163]. Both hepatocytes and Kupffer cells are key sources of ROS production, but Kupffer cells have a greater impact in aggravating inflammation in the liver [164].

Oxidative stress polarizes Kupffer cells toward an M1 pro-inflammatory phenotype [177]. These activated Kupffer cells release cytokines, chemokines, nitric oxide, and ROS. These signals help in the recruitment of leukocytes such as neutrophils, monocytes, natural killer cells, and natural killer T (NKT) cells, which further supports inflammation in the liver [178,179].

In response to lipotoxicity, enhanced formation of ROS in hepatocytes contributes to the production of pro-inflammatory cytokines mediated by stress signaling pathways including c-Jun N-terminal kinases 1/2 (JNK1/2) and STAT1 [175,180]. Grohmann et al. showed oxidative stress-mediated activation of STAT1 signaling in the liver in obese mice and humans with MASLD and infiltration of lymphocytes into the liver by increased production of the lymphocyte-specific chemokine CXCL9 [180]. Another connection linking oxidative stress with hepatic inflammation is the activation of the NLRP3 inflammasome by ROS which further leads to caspase 1-mediated release of active IL-1β [181,182,183]. Moreover, mitochondrial DNA released from damaged hepatocytes during oxidative stress-mediated mitochondrial dysfunction causes liver inflammation by activation of TLR-9 [184].

Oxidative stress also activates NF-κB and activating protein-1 (AP-1) transcription factors, leading to an inflammatory response and activation of cell death pathways in injured hepatocytes [185]. NF-κB provokes the transcription of genes involved in triggering inflammatory responses [186]. TNF-α and IL-1 induce production of ROS, which further activates the NF-κB pathway in MASLD [187]. It has been found that nuclear factor erythroid 2-related factor 2 (NRF2), which negatively regulates the NF-κB pathway by inhibiting nuclear translocation of NF-κB and blocking the degradation of IκB-α, may be a viable treatment target in slowing MASLD progression [188,189].

NADPH oxidase (NOX) is the major group of enzymes that contribute to ROS production in the liver. NOX isoforms NOX1, NOX2, and NOX4 activate Kupffer cells and thus have a significant role in the inflammatory response [190,191]. NOX2-generated ROS enhance the production of pro-inflammatory cytokines such as TNFα, IL-6, and IL-1β. These pro-inflammatory cytokines help in the infiltration of neutrophils, which further induce activation of HSCs [144,154,192]. NOX-mediated ROS production is coupled with TGF-β signaling, leading to liver pathologies such as liver fibrosis and cancer [193]. In HSCs, TGF-β increases NOX expression and ROS production, and ROS can also induce activation of TGF-β [194,195]. TGF-β activates HSCs via the SMAD pathway leading to the overproduction of matrix proteins and fibrogenesis [196,197,198].

Figure 2 presents an overview of the pathways, processes, and cell types involved in oxidative stress as a central influence on the development of MASLD.

## 5. Treatments

The AACE 2013 and The European Association for the Study of Obesity have recommended using the term “adiposity-based chronic disease” (ABCD) as a medical diagnosis for obesity. This approach emphasizes treating ABCD to prevent the progression of NAFLD and NASH [199].

### 5.1. Lifestyle Intervention Is the Primary Treatment for Pediatric MASLD

In MASLD, liver fat accumulation results from an imbalance between lipid deposition and removal, driven by hepatic synthesis of triglycerides and de novo lipogenesis. Weight loss is the most effective way to promote liver fat removal because it decreases the delivery of free fatty acids to the liver, increases extrahepatic insulin sensitivity, and reduces adipose tissue inflammation [200,201,202]. Intensive lifestyle intervention with dietary and physical activity modification supported by behavioral change strategies is the mainstay of therapy for pediatric MASLD, best carried out by a multidisciplinary team generally composed of a medical provider, registered dietician, exercise specialist, mental health professional, nurse, and social worker [203,204].

Lefere et al. reported a prospective study of 204 children and adolescents (aged 8–18 years old) undergoing intensive lifestyle therapy for severe obesity at a tertiary center in Belgium [205]. Intensive lifestyle therapy included caloric restriction, physical activity, education on a healthy lifestyle, and psychosocial support. At baseline and 6 and 12 months, liver ultrasound and transient elastography with controlled attenuation parameters were performed to assess liver steatosis and fibrosis. MASLD on ultrasound was present in 71.1%, and 32.8% of patients had fibrosis. After 6 months, lifestyle intervention resulted in 16% median body weight loss in 167 patients, and fibrosis improved by 75.0% (*p* < 0.001). Seventy-nine patients had reached the 1-year time point. The improvements were sustained because fibrosis regressed by at least 1 stage in all patients with baseline fibrosis. Fasting serum ALT and homeostasis model assessment of insulin resistance decreased significantly over the 1-year period (*p* < 0.001). MASLD and associated fibrosis are highly prevalent in children and adolescents with severe obesity. An intensive, multidisciplinary lifestyle management program that causes significant weight loss improves not only liver steatosis but also fibrosis.

A randomized control trial of 139 children, aged 8–17 years old, with MASLD receiving lifestyle advice was conducted by Xanthakos et al. [206]. Over a mean period of 1.6 ± 0.4 years, they showed that dietary and physical activity recommendations led to MASH resolution in 29% and fibrosis improvement in 34% of children. However, in the same timeframe, 18% of children progressed to definite MASH, and 23% had worse fibrosis. Increasing gamma-glutamyl transferase was the only factor linked to lower odds of improvement (MASH resolution and/or fibrosis regression). Identifying these risk factors may help find children who are not responding to lifestyle counseling and require more intensive interventions.

Evidence supporting the effect of exercise training on the reduction in hepatic fat content and MASLD prevalence is limited in the pediatric population. A meta-analysis on overweight and obese children and adolescents showed that supervised exercise training programs significantly reduced the intrahepatic fat percentage [207]. Documenting exercise history can inform the design of an exercise program [208]. Combining aerobic and resistance exercise is recommended [209].

A systematic review and meta-analysis were conducted to define an optimal supervised exercise training regimen describing the type (e.g., aerobic, resistance, or combined exercise training), intensity (e.g., moderate or vigorous), volume, and frequency of exercise needed to reduce hepatic fat content in children and adolescents [210]. Supervised exercise training interventions performed in children and adolescents (6–19 years) significantly reduced hepatic fat content compared to the control groups. This systematic review and meta-analysis showed that supervised exercise training could be an effective strategy in managing and preventing MASLD in children and adolescents 6–19 years old. Both aerobic and resistance training, at vigorous or moderate-to-vigorous intensities, with sessions lasting 60 min or more and a frequency of three or more sessions per week, aimed at improving cardiorespiratory fitness and muscular strength, were beneficial for reducing hepatic fat content in youth. These data concur with the international recommendations of physical activity for health promotion in youth. They may be useful when designing exercise training programs to improve and prevent hepatic steatosis in the pediatric population.

The currently recommended management of MASLD in children and adolescents emphasizes weight loss through lifestyle modification, including caloric restriction and increased physical activities. These conventional treatments aim to reduce weight, but the process can be lengthy and may lead to ongoing liver injury [211]. Care of children with MASLD includes a recommendation of lifestyle changes, promotion of dietary changes to create an energy deficit, with a reduction in sugar consumption as the first-line lifestyle modification, and encouragement of increased physical activity aiming for BMI optimization.

### 5.2. Medical Therapy

At this time, no licensed or uniformly recommended pharmacological therapies exist for MASLD [212]. Among adults with MASH, weight loss of >5% total body weight can reduce hepatic steatosis, weight loss of >7% of total body weight can improve MASH, and weight loss of >10% of total body weight can result in fibrosis regression/stability [213]. Although improvement in BMI z-score was proportionally associated with resolution of MASH in one prospective study, for children failing to respond to standard lifestyle counseling, multidisciplinary interventions can be more effective, particularly with increased contact hours and additional behavioral counseling [206,214]. These resource-intensive multidisciplinary programs are often less available, affordable, or accessible. They are also mostly ineffective for severely obese children, who made up a significant proportion of the cohort (25% with BMI z-score ≥ 2.5).

Pharmacotherapy as an adjunctive therapy to lifestyle modification for children and adolescents with obesity and MASLD or MASH is advised when lifestyle modification alone is not associated with improved or resolution of MASH and fibrosis. However, at present, there is a shortage of medications available for treating pediatric MASLD. There are no FDA-approved pharmacotherapies for pediatric MASLD, and bariatric surgery is only recommended if additional comorbidities exist [212].

#### 5.2.1. Metformin

As discussed earlier, MASLD is regarded as a hepatic manifestation of metabolic syndrome, and obesity, insulin resistance, dyslipidemia, abnormal glucose metabolism, and hypertension are risk factors for MASLD [215]. Therefore, improving insulin resistance and adjusting the balance of glucose and lipid metabolism may be an important measure for preventing and treating MASLD. Metformin is an insulin sensitizer that may improve insulin sensitivity by increasing the binding of peripheral insulin and insulin receptors, increasing the clearance of blood sugar, and improving insulin sensitivity [216]. Additionally, the gastrointestinal discomfort caused by metformin can decrease food intake and body weight [217]. Nevertheless, metformin’s exact mode of action in liver steatosis has not been confirmed or fully explored [218].

Experimental studies indicate that metformin exerts its anti-steatotic effects primarily through the upregulation of specific proteins that play crucial roles in mitochondrial biogenesis and oxidative metabolism [219,220]. Key proteins involved in these processes include peroxisome proliferator-activated receptor gamma coactivator 1α (PGC-1α) and peroxisome proliferator-activated receptor α (PPARα), which enhance the formation and proliferation of mitochondria. Additionally, proteins such as cytochrome c oxidase subunit IV (COX IV), cytochrome c, and hydroxyacyl-CoA dehydrogenase short chain (HADHSC) are also upregulated, facilitating the oxidation of free fatty acids. This coordinated action not only aids in the breakdown of fatty acids but also contributes to improved energy metabolism, ultimately helping to combat excess fat accumulation in the liver [221]. Furthermore, metformin positively impacts MASLD by activating adenosine monophosphate-activated protein kinase (AMPK), which regulates glucose and lipid metabolism [222,223]. Another hypothesis is that oral administration of metformin is associated with changes in the gut microbiota, which reduces the translocation of bacterial endotoxins, resulting in improved insulin resistance in patients with MASLD [224,225].

Recent studies have examined the use of metformin in children with MASLD and showed promising results, but they are still controversial [226,227,228]. A systemic review of pediatric MASLD patients treated with metformin demonstrated a decreased steatosis on ultrasound and improved insulin resistance, which may benefit liver histology [229]. Moreover, a meta-analysis conducted by systematic literature search through major electronic databases investigated metformin’s efficacy and safety in pediatric MASLD patients until 12 March 2023. Four randomized controlled trials (RCTs) with 309 pediatric patients with MASLD were included in the meta-analysis. In this meta-analysis, metformin failed to improve liver enzymes statistically; however, it may be beneficial in improving lipid parameters and insulin metabolism regulation in pediatric patients with MASLD. As there were not enough available studies in the literature, the influence of metformin on liver ultrasonography or histology in pediatric MASLD was not evaluated [230]. Of note is that the results of a recent meta-analysis in adult patients with MASH showed no improvement in liver histology parameters in the group treated with metformin [231].

#### 5.2.2. GLP-1 Receptor Agonists

Increased understanding of the role of gastrointestinal hormones in signaling hunger, satiety, and energy homeostasis has led to progress in available therapies [232]. Appetite-related gut hormones have become important targets for developing pharmacologic treatments for obesity and its comorbidities [233].

GLP-1 is the incretin hormone released by the ileal L-cells after food consumption, inducing satiety by delaying gastric emptying and glucagon secretion. It also controls appetite and satiety in the hypothalamus [214,234]. GLP-1 receptor agonists have become essential components of pharmacotherapy for obesity and type 2 diabetes due to their significant clinical benefits, which include weight loss, improved glycemic control, and enhancements in cardiometabolic health. This is achieved through decreased gastrointestinal motility and an anorectic effect, activating central GLP-1 receptors in the brain, especially in the arcuate nucleus region [235]. The favorable metabolic profile induced by GLP-1 receptor agonists (weight loss and reduced caloric intake, improvement in glycemic compensation) allows one to hypothesize their potential effectiveness in treating MASLD, in addition to type 2 diabetes and obesity. GLP-1 receptor agonists positively impacted steatosis and liver inflammation, and potentially fibrosis degree. To date, it is unknown if the benefits of treatment with GLP-1 receptor agonists are direct or mediated by drug-induced weight loss [236,237].

The first GLP-1 receptor agonist to be approved for preadolescent and adolescent populations aged 10 to 17 was liraglutide in 2019 for treating type 2 diabetes. Soon after, semaglutide was also approved for preadolescents and adolescents aged 12 to 17 years of age with obesity [238].

GLP-1 receptor agonists are not just effective for weight loss; they also play a crucial role in reducing lipotoxicity [239]. By promoting enhanced insulin sensitivity in the liver, they can boost mitochondrial function, leading to even greater health benefits.

Semaglutide offers superior weight loss outcomes than other GLP-1 receptor agonists due to its unique ability to control appetite, reduce food cravings, and decrease the desire for fatty foods [240]. Its metabolism primarily occurs through the enzyme neprilysin, which leads to higher plasma levels than liraglutide, thereby contributing to its more significant anti-obesity effect [241]. Additionally, semaglutide has a greater affinity for the GLP-1 receptor, which further enhances its efficacy on GLP-1 receptors. Weekly administration of semaglutide enhances patient adherence compared to the daily doses of liraglutide [242]. In adolescents, a significant decrease in BMI has been observed after 3 months of use [243]. However, there are no randomized control trials of a GLP-1 receptor agonist for the treatment of MASLD in children and adolescents with MASH. A recent small retrospective case series by Choi et al., including nine obese children older than 10 with type 2 diabetes or prediabetes treated with GLP-1 receptor agonists for 12 months, revealed an ALT decrease of an average of 98 points, hemoglobin A1c decrease of an average of 2.2 points, and BMI decrease of an average of 2.4 points [244].

#### 5.2.3. Resmetirom

Patients with MASH have an impaired function of liver-directed thyroid hormone receptor-β, resulting in decreased mitochondrial function and β-oxidation of fatty acids, along with an increase in fibrosis [245]. Resmetirom is an oral, liver-directed thyroid hormone receptor-β-selective agonist recently approved for the treatment of MASH in adults with moderate to advanced fibrosis. MASH resolution with no worsening of fibrosis, and an improvement in fibrosis by ≥1 stage with no worsening of the NAFLD activity score, was achieved in significantly more patients who received resmetirom than in those who received placebo at week 52 [246]. In addition, the levels of a various range of atherogenic lipids and lipoproteins were shown to be reduced by resmetirom compared to placebo, consistent with earlier studies [247]. Diarrhea and nausea were the most common adverse events reported and were generally self-limited. However, the long-term safety of this medication has not yet been evaluated.

The approval of resmetirom has created new opportunities for MASH drug development and has offered essential insights into future clinical trial designs and treatment strategies. This represents a breakthrough for adult patients with MASH, also inspiring hope for novel therapeutic options in the pediatric population in the future.

#### 5.2.4. Antioxidants

Vitamins play a crucial role in regulating key enzymatic processes in the liver, and changes in vitamin metabolism have been shown to significantly influence the progression of MASLD [248,249].

Vitamin E is a fat-soluble vitamin and potent antioxidant that effectively neutralizes ROS and nitrogen species and enhances the activity of antioxidative enzymes [250]. It is consumed in the diet through plant-based oils. Therefore, vitamin E has been extensively studied in MASLD. Supplementary vitamin E has been found to improve MASH in randomized clinical trials [251,252,253]. Vitamin E supplementation was recommended in patients with proven MASH without diabetes by the American Association for the Study of Liver Disease [254].

The antioxidant properties of vitamin E primarily stem from the hydroxyl group on the aromatic ring of tocopherols and tocotrienols (known collectively as tocochromanols), which act as a hydrogen donor capable of neutralizing free radicals or ROS [255]. Vitamin E has been used as monotherapy or in combination with other agents for treating MASLD or MASH in various clinical trials with duration ranging from 24 weeks to over 2 years. These studies have reported both biochemical and histological improvements in the liver parameters, including steatosis, lobular inflammation, balloon degeneration, and fibrosis [256].

A clinical trial from Nobili et al. used vitamin E in combination with another antioxidant, hydroxytyrosol, and found that the treatment was well tolerated by children with MASLD and may improve oxidative stress parameters, insulin resistance, and steatosis [138]. A subset of children with MASLD from the Nobili study were followed for 2 years after the end of the 4-month treatment period, and decreased steatosis was seen in the group originally treated with vitamin E and hydroxytyrosol compared to those given a placebo [257].

The TONIC trial compared the effect of placebo, metformin, and Vitamin E in children diagnosed with histologically confirmed MASLD. The primary outcome measured was a sustained reduction in ALT levels. The results showed that neither metformin nor Vitamin E significantly improved over placebo regarding the primary outcome. However, both treatment groups demonstrated evidence of reduced hepatocyte injury on biopsy [218].

#### 5.2.5. Vitamin D

Vitamin D is a fat-soluble steroid vitamin that plays an important role in the function of many organs including the heart and liver. It can be synthesized with skin exposure to ultraviolet light or consumed in the diet [258]. Research suggests that low vitamin D levels may be linked to obesity, inflammation, insulin resistance, dyslipidemia, and coronary artery disease [259,260,261]. In addition, it has been demonstrated that vitamin D regulates insulin secretion by pancreatic β cells [262]. Furthermore, vitamin D receptors exist in hepatic cells, and vitamin D deficiency leads to increased inflammatory processes in chronic liver diseases [263,264]. The activation of vitamin D receptors increases insulin sensitivity, and vitamin D transcriptionally activates the human insulin gene [265,266,267]. A meta-analysis by Guo et al. assessing the effect of vitamin D on insulin resistance provides substantial evidence that vitamin D has a favorable effect on insulin sensitivity [244]. Mirhosseini et al. found a salutary effect of vitamin D on insulin sensitivity in persons with prediabetes [268]. Another recent systematic review and meta-analysis with an updated literature review involving more randomized clinical trials provided stronger evidence that vitamin D supplementation improves insulin resistance in patients with MASLD [269]. When the improvement of insulin resistance associated with vitamin D supplementation was analyzed in terms of liver function, as indicated by changes in ALT and AST levels, the results showed a decrease in ALT levels with additional vitamin D supplementation, while no significant change was observed in AST levels. It should be noted that not all studies have found an effect of vitamin D on insulin sensitivity. A systematic review and meta-analysis conducted by Pramono et al., which included 18 randomized clinical trials, evaluated the impact of vitamin D supplementation on insulin sensitivity in individuals with or at risk for insulin resistance [270]. The findings revealed that additional vitamin D treatment had no effect on insulin sensitivity.

In children and adolescents, a randomized controlled study was conducted in 109 youths below the age of 18 years with biopsy-proven MASLD. The participants were randomly divided into two groups: the treatment group, which received 2000 IU/day vitamin D for 6 months, and the control group, who received a placebo. Out of the initial participants, 100 patients completed the study. The results indicated that vitamin D supplementation significantly reduced hepatic steatosis and lobular inflammation, resulting in improvements in the grades of MASLD in young persons, as confirmed by liver biopsy. However, no effects on hepatocyte ballooning or fibrosis were observed. Therefore, adjuvant vitamin D supplementation is recommended in children with MASLD [271].

#### 5.2.6. Prebiotics and Probiotics

Changes in the gut–liver axis and shifts in the gut microbiome are significant risk factors for the development of MASLD [272,273]. Patients with this condition exhibit increased bacterial overgrowth in the small intestine and have impaired intestinal permeability [274].

Probiotics are live microorganisms that can provide health benefits to the host [275]. The primary genera of probiotics that have been extensively studied are Lactobacillus and Bifidobacterium. In contrast, prebiotics are non-viable food components that help modulate the microbiota and can also confer health benefits to the host. Prebiotics mainly consist of polysaccharides such as inulin, cellulose, hemicellulose, pectins, and resistant starch, as well as oligosaccharides like fructooligosaccharides, galactooligosaccharides, isomaltooligosaccharides, xylooligosaccharides, lactulose, and soy oligosaccharides. These components stimulate the growth of beneficial bacteria. Among the prebiotics studied in patients with MASLD, fructooligosaccharides are the most researched [276]. Lastly, synbiotics refer to a combination of probiotics and prebiotics [277].

Probiotics positively affect inflammatory liver damage through regulation of c-Jun N-terminal kinase (JNK) and nuclear factor kappa B (NF-κB), which is associated with regulation of tumor necrosis factor-alpha (TNF-α) and insulin resistance [278].

Prebiotics can selectively enhance the growth and activity of intestinal microbes [279]. Research using animal models has demonstrated that prebiotic supplementation can reduce the fatty acid synthesis pathway, potentially decreasing the accumulation of fructose-induced hepatic triglyceride [280]. This effect may be linked to a reduction in the gene expression of enzymes that regulate hepatic lipogenesis, such as acetyl Co-A carboxylase and fatty acid synthase [281]. Additionally, oligofructose alters the composition of intestinal microbiota, promoting the growth of Bifidobacterium, which in turn improves mucosal barrier function and lowers endotoxin levels [282].

### 5.3. Surgical Intervention

Bariatric surgery is a recognized treatment option for young individuals with class II severe obesity (defined as having a body mass index (BMI) between 35 kg/m^2^ and 40 kg/m^2^ or being at least 120% to less than 140% of the 95th percentile for their age) who also have significant comorbidities [283]. It is also appropriate for those with class III obesity, which includes individuals with a BMI of 40 kg/m^2^ or greater, regardless of whether they have accompanying health conditions. It is an accepted treatment for youth with class II severe obesity (≥120% to <140% of the 95th percentile or BMI ≥ 35 kg/m^2^) with significant comorbid conditions such as hypertension, hyperlipidemia, diabetes, sleep apnea, poor self-esteem, and even serious forms of depression and for those with class III with or without comorbid conditions (≥140% of the 95th percentile or BMI ≥ 40 kg/m^2^) [284,285]. The surgical risks for adolescents are similar to those for adults, but they may encounter long-term nutritional complications secondary to a significant decrease in adherence to vitamin and mineral supplementation regimens within the first few months after surgery [286]. While bariatric surgery can improve histological outcomes in pediatric patients with MASH and reduce liver fibrosis in obese adolescents at one year post-surgery, its use as a treatment option remains controversial in all but exceptional clinical situations [287,288].

In summary, managing MASLD in the pediatric population involves preventing and treating obesity from a very young age to reduce the disease burden of obesity and obesity-related comorbidities. Based on the current literature, we recommend a comprehensive, family-based, multi-disciplinary behavioral intervention that focuses on lifestyle changes. This includes a calorie-controlled balanced diet, regular vigorous physical activity and exercise, a reduction in sedentary habits, and support from the entire family, school, and community. Current evidence regarding the effectiveness and safety of most pharmacotherapy and bariatric surgery treatments for adolescents is still limited, and there are insufficient data for children.

## 6. Conclusions

Childhood obesity is a complex, multifactorial disease that greatly impacts the liver and also many other vital organ systems. Body weight in the early years of life is influenced by lifestyle, environmental and social factors, genetics, birth history, and maternal behaviors. Pediatric MASLD has increased with the rising prevalence of pediatric obesity and is considered a metabolic disruption emerging from various combinations of high energy intake, obesity, sedentary lifestyle, insulin resistance, and type 2 diabetes. The true prevalence of pediatric MASLD in obesity remains unknown. Optimization of screening practices will help in determining an accurate estimation of the health and socioeconomic burden of MASLD starting in childhood. Timely investigation and informative counseling of patients early on regarding this chronic disease, including hepatic and extra-hepatic risks, is paramount. Current data do not allow us to predict the progression and natural history of pediatric MASLD accurately. To address this complex issue, future studies designed to incorporate a very large sample size, diversity in age, gender, race, and geographic location, and consistent follow-up histology are needed. A standardized pathology evaluation and thorough phenotyping assessments of BMI, liver transaminases, and MASLD-related comorbidities at each biopsy time point would be ideal. There is currently no simple method for treating MASLD. Ongoing research into the complex pathophysiology of MASLD is uncovering potential therapeutic targets. While the search for effective treatments continues, healthy eating and physical activity remain the proven prevention and treatment strategies for pediatric MASLD.

## Figures and Tables

**Figure 1 metabolites-15-00287-f001:**
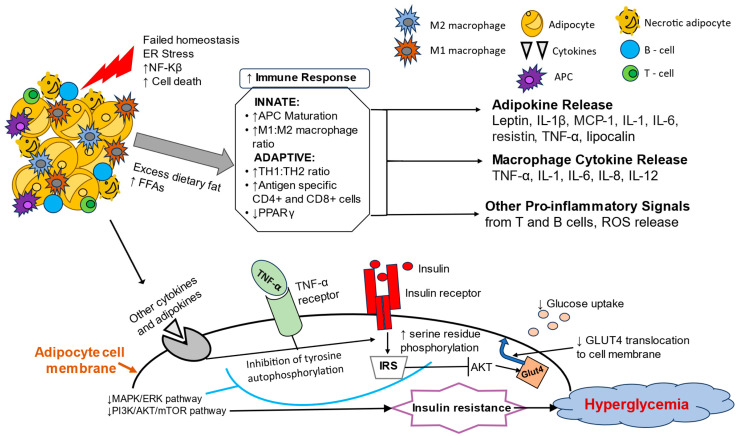
Mechanism of both systemic inflammation and insulin resistance triggered by visceral obesity. An overload of lipids leads to the expansion of adipocytes in obese visceral adipose tissue. Excess lipids cause adipocyte dysfunction, and the adipose tissue can then no longer maintain homeostasis causing endoplasmic reticulum (ER) stress and increased expression of inflammatory signals, such as nuclear factor kappa β (NF-kβ), resulting in adipocyte death. Necrotic adipose tissue attracts macrophages and other immune cells (B cells, T cells, antigen-presenting cells [APCs]). Prolonged overnutrition and excess dietary fat promote free fatty acid release from adipose tissue, in turn activating the innate and adaptive immune responses. This immune activation manifests as chronic low-grade inflammation. Adipose tissue inflammation significantly elevates the levels of pro-inflammatory adipokines, macrophage cytokines, and other pro-inflammatory signals such as reactive oxygen species (ROS). This results in interference with insulin signaling, disrupting the cellular localization and phosphorylation state of constituents. TNF-α activation of the TNF-α receptor and other cytokines/adipokines inhibit tyrosine autophosphorylation and lead to phosphorylation of serine residues on insulin receptor substrate (IRS) proteins. When serine residues on the IRS are phosphorylated, protein kinases such as protein kinase B (AKT) are blocked, causing decreased glucose uptake via the glucose transporter type 4 (GLUT4). Additionally, there is disrupted downstream insulin signaling via the mitogen-activated protein kinase/extracellular signal-regulated kinase (MAPK/ERK) and phosphoinositide 3-kinase/protein kinase B/mammalian target of rapamycin (PI3K/AKT/mTOR) pathways also contributing to insulin resistance. The subsequent hyperglycemia may further amplify the inflammatory response as well and add to the ongoing systemic inflammation. Additional abbreviations: IL—interleukin; MCP-1—monocyte chemoattractant protein-1; PPARγ—peroxisome proliferator-activated receptorγ; ↑—increased; ↓—decreased.

**Figure 2 metabolites-15-00287-f002:**
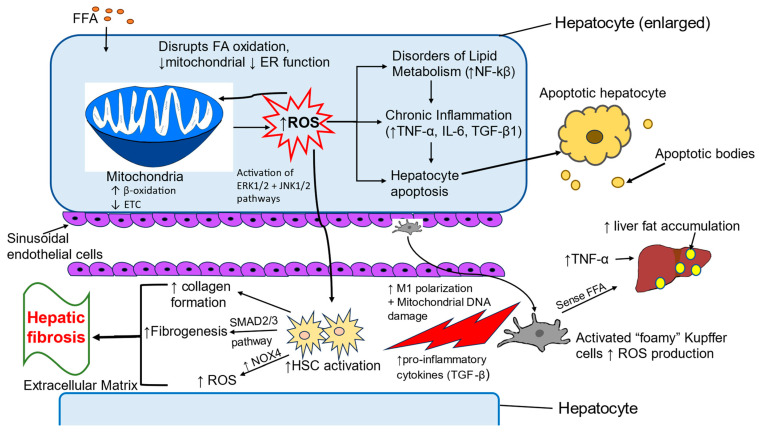
Lipotoxicity in the liver and the mechanism of progression to metabolic dysfunction-associated steatotic liver disease (MASLD). Influx of free fatty acids (FFAs) results in hepatocyte enlargement and disruption of fatty acid (FA) oxidation homeostasis. Increased mitochondrial beta-oxidation and decreased electron transport chain (ETC) activity along with endoplasmic reticulum (ER) dysfunction contribute to the accumulation of reactive oxygen species (ROS) within hepatocytes. ROS accumulation results in disordered lipid metabolism via production of inflammatory signals such as nuclear factor kappa β (NF-kβ) and chronic inflammation via tumor necrosis factor-α (TNF-α), interleukin-6 (IL-6), and transforming growth factor-β1 (TGF-β1). Once hepatocytes can no longer maintain homeostasis, apoptosis ensues and triggers the extracellular signal-related kinase 1/2 (ERK1/2) and c-Jun N-terminal kinase 1/2 (JNK1/2) pathways. Both pathways contribute to the activation of hepatic stellate cells (HSCs) and infiltration of Kupffer cells. As fatty acids continue to accumulate during MASLD, Kupffer cells undergo M1 polarization, becoming activated “foamy” Kupffer cells. Activated foamy Kupffer cells can sense FFA accumulation and produce TNF-α, further contributing to liver fat accumulation. Activated Kupffer cells also elicit the increased production of ROS and pro-inflammatory cytokines. This process further activates HSC, prompting collagen formation, fibrogenesis via the SMAD2/3 pathway, and increased ROS production via NADPH oxidase 4 (NOX4), ultimately causing progression of MASLD and eventually hepatic fibrosis. ↑—increased; ↓—decreased.

**Table 1 metabolites-15-00287-t001:** Etiology of obesity in children.

Cause	Examples	References
Exogenous	Excessive caloric intake, sedentary lifestyle, insufficient sleep, increased screen time, parenting styles, lack of school resources, community factors, ethnicity	[18,20,31,32,33,34]
Monogenic	Leptin deficiency, leptin receptor gene deficiency, POMC deficiency, MC4R gene mutation, PCSK1 deficiency, SIM 1 deficiency, KSR2 deficiency, NTRK2 mutations, BDNF mutations	[53]
Syndromes	Albright hereditary osteodystrophy, Prader–Willi syndrome, Down syndrome, Turner syndrome, Bardet–Biedl syndrome, WAGR syndrome, Fragile X syndrome, Cohen syndrome, Beckwith–Wiedemann syndrome	[50,51,52,53]
Endocrine	Hypothyroidism, Cushing syndrome, growth hormone deficiency, hyperinsulinism, PCOS	[18,35]
Medications	Glucocorticoids, antipsychotics, beta blockers, lithium, antidepressants, antiepileptics, diabetes medications	[18,35]
Neurologic	Traumatic brain injury, neoplasms, post-cranial radiation, hypothalamic obesity	[35,54,55]
Psychiatric	Depression, eating disorders	[18,20,35]

Abbreviations: POMC—proopiomelanocortin; MC4R—melanocortin 4 receptor; PCSK1—procon-vertase 1; SIM1—single-minded homolog 1; KSR2—kinase suppressor of RAS 2; NTRK2—neurotrophic tyrosine kinase receptor type 2; BDNF—brain-derived neurotrophic factor; PCOS—polycystic ovarian syndrome; WAGR—Wilms tumor, aniridia, genitourinary malformations and a range of mental disabilities.

## Data Availability

No new data were created or analyzed in this study. Data sharing is not applicable to this article.

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
