# Peer review of "From Childhood Obesity to Metabolic Dysfunction-Associated Steatotic Liver Disease (MASLD) and Hyperlipidemia Through Oxidative Stress During Childhood"

_metabolites, 2025, doi:10.3390/metabo15050287_

Round 1

Reviewer 1 Report

Comments and Suggestions for Authors

This is a very comprehensive review on pediatric obesity and metabolic dysfunction-associated steatotic liver disease MASLD and hyperlipidemia through oxidative stress.  It is well written to the eyes of a non native speaker. It has compiled in this review the major advancements in comprehension and treatment of childhood obesity and MASLD which is very much needed in the filed.

I have however a few comments I would like to see adressed:

The title suggests that this review will follow a roadmap "From" childhood obesity "to" MASLD and hyperlipidemia through oxidative stress (in adulthood). However, the mais focus is pediatric, specially treatment. I would rephrase it as From obesity to metabolic dysfunction-associated 2 steatotic liver disease (MASLD) and hyperlipidemia through 3 oxidative stress during childhood. This reflects more accurately the nature of this review

At line 47, authors refer to "MASLD has become a major public health problem in adults and children in industrialized countries". This is a common reference in the same line as "developed countries", but I believe it is important to actually point ou that this is a problem in countries with high ultraprocessed food consumption and sedentarism, so the real cause is made more evident. Although this is adressed in Etiology section, it should be pointed out here as well.

At line 116/117, authors stated "Reduced physical activity intensified during the 116 COVID-19 pandemic and showed a rapid increase in the onset of pediatric obesity [45,46]." As we all know COVID-19 had a huge impact worldwide, it deserves at least 1 paragraph adressing its impact in pediatric obesity

In figure 1, Insulin signaling is represented by IRS acting through AKT, which signals Glut4 translocation. This is true for skeletal muscle and adipose tissue, however, in the figure, it is not clear whether the lower panel is an enlargement of an adipocyte membrane or systemic action. In case of systemic action, Glut4 translocation would be misleading, as it only occurs on muscle and fat.

Section 3.1 Hormones and Obesity should be Section 2.4 and section 2.4 Associated Comorbidities of Childhood Obesity changed to 2.5. So section 3.1 would be section 3.2 Definition of metabolic dysfunction- associated steatotic liver disease MASLD. and so on (This makes a lot more sense to me).

Lines 647/648 seems off. Maybe it would be better placed to star section 5.2

Author Response

We thank the reviewer for thoroughly scrutinizing our manuscript. As requested, we have revised the manuscript and addressed the specific comments of the reviewer. The revised sections are delineated in red in a marked copy of the manuscript text.

Below, we provide a point-by-point response to the reviewer’s comments.

Reviewer # 1 Comments and Responses

  • COMMENT #1: The title suggests that this review will follow a roadmap "From" childhood obesity "to" MASLD and hyperlipidemia through oxidative stress (in adulthood). However, the main focus is pediatric, specially treatment. I would rephrase it as From obesity to metabolic dysfunction-associated 2 steatotic liver disease (MASLD) and hyperlipidemia through oxidative stress during childhood. This reflects more accurately the nature of this review.

RESPONSE: We have changed the title as recommended.

  • COMMENT #2: At line 47, authors refer to "MASLD has become a major public health problem in adults and children in industrialized countries". This is a common reference in the same line as "developed countries", but I believe it is important to actually point out that this is a problem in countries with high ultraprocessed food consumption and sedentarism, so the real cause is made more evident. Although this is addressed in Etiology section, it should be pointed out here as well.

RESPONSE: We have made this point as suggested (lines 48-49).

  • COMMENT #3: At line 116/117, authors stated "Reduced physical activity intensified during the COVID-19 pandemic and showed a rapid increase in the onset of pediatric obesity [45,46]." As we all know COVID-19 had a huge impact worldwide, it deserves at least 1 paragraph addressing its impact in pediatric obesity.

RESPONSE: We have added this paragraph with accompanying references (lines 119-131).

  • COMMENT #4: In figure 1, Insulin signaling is represented by IRS acting through AKT, which signals Glut4 translocation. This is true for skeletal muscle and adipose tissue, however, in the figure, it is not clear whether the lower panel is an enlargement of an adipocyte membrane or systemic action. In case of systemic action, Glut4 translocation would be misleading, as it only occurs on muscle and fat.

RESPONSE: We have modified figure 1 and labeled the adipocyte.

  • COMMENT #5: Section 3.1 Hormones and Obesity should be Section 2.4 and section 2.4 Associated Comorbidities of Childhood Obesity changed to 2.5. So section 3.1 would be section 3.2 Definition of metabolic dysfunction- associated steatotic liver disease MASLD. and so on (This makes a lot more sense to me).

RESPONSE: We have moved the sections.

  • COMMENT #6: Lines 647/648 seems off. Maybe it would be better placed to star section 5.2

RESPONSE: We have moved lines 647/648 to section 5.2.

           We thank the reviewer and believe that the manuscript is improved as a result of their input.  We hope you will agree and decide in favor of accepting our report at this time.

Reviewer 2 Report

Comments and Suggestions for Authors

In this review, the authors aimed to address the common factors relating to childhood obesity, oxidative stress, Insulin resistance, and MASLD in the pediatric population.

Topics that are discussed include the role of insulin resistance and oxidative stress in causing the alterations in the hepatic structure characteristic of MASLD and the therapeutic approaches in the pediatric population.

The manuscript is well presented and discussed. I have a minor comment to further improve this review. Discussing the causes of non-alcoholic steatotic liver disease, the authors should also take into account the role of intestinal permeability. Considering the growing prevalence in the western countries of celiac disease, the authors should recall also the role of dietary intervention and the role of gluten-free diet in patients with celiac disease where it has been demonstrated that gluten-free diet initiation is significantly associated with liver steatosis development, with a prevalence up to 34%. Importantly, the relative risk for NAFLD was higher in non-overweight celiac disease patients, as previously demonstrated (doi: 10.1111/apt.14910.). This suggest that dietary advice with a patient-tailored approach should assist celiac disease patients starting gluten-free diet in achieving an appropriate nutritional intake whilst reducing the risk of long-term liver-related events. This should be recalled and discussed.

Author Response

We thank the reviewer for thoroughly scrutinizing our manuscript. As requested, we have revised the manuscript and addressed the specific comments of the reviewer. The revised sections are delineated in red in a marked copy of the manuscript text.

Below, we provide a point-by-point response to the reviewer’s comments.

Reviewer # 2 Comments and Responses

  • COMMENT #1: Discussing the causes of non-alcoholic steatotic liver disease, the authors should also take into account the role of intestinal permeability. Considering the growing prevalence in the western countries of celiac disease, the authors should recall also the role of dietary intervention and the role of gluten-free diet in patients with celiac disease where it has been demonstrated that gluten-free diet initiation is significantly associated with liver steatosis development, with a prevalence up to 34%. Importantly, the relative risk for NAFLD was higher in non-overweight celiac disease patients, as previously demonstrated (doi: 10.1111/apt.14910.). This suggests that dietary advice with a patient-tailored approach should assist celiac disease patients starting gluten-free diet in achieving an appropriate nutritional intake whilst reducing the risk of long-term liver-related events. This should be recalled and discussed.
  • RESPONSE: We appreciate this suggestion and have now added this content to the manuscript with corresponding references (lines 435-459).

We thank the reviewer and believe that the manuscript is improved as a result of their input.  We hope you will agree and decide in favor of accepting our report at this time.

Reviewer 3 Report

Comments and Suggestions for Authors

Reviewer Comments

  1. There are numerous abbreviations and terms used that are not defined in the Materials and Methods section. Please clarify their purpose and what they are used to evaluate, including CD11c, TNF-α, IRS-1 and others.
  2. Each study is discussed individually without integrating them into a broader perspective in the treatment section.
  3. Numerous factors can influence the treatment, particularly exercise history. These should be acknowledged in the paper.
  4. Conduct a fibrosis regression analysis to account for confounding variables such as age, gender, race and family medical history.
  5. The most common factors increase the risks of fractures, slipped capital femoral epiphysis, and joint pain. Providing more details on these risks would benefit the reader.

Author Response

We thank the reviewer for thoroughly scrutinizing our manuscript. As requested, we have revised the manuscript and addressed the specific comments of the reviewer. The revised sections are delineated in red in a marked copy of the manuscript text.

Below, we provide a point-by-point response to the reviewer’s comments.

Reviewer # 3 Comments and Responses

  • COMMENT #1: There are numerous abbreviations and terms used that are not defined in the Materials and Methods section. Please clarify their purpose and what they are used to evaluate, including CD11c, TNF-α, IRS-1 and others.

     RESPONSE: We have defined and clarified these abbreviations.

  • COMMENT #2: Each study is discussed individually without integrating them into a broader perspective in the treatment section.

     RESPONSE: We have added a paragraph integrating the various studies into a broader perspective at the end of this section.

  • COMMENT #3: Numerous factors can influence the treatment, particularly exercise history. These should be acknowledged in the paper.

     RESPONSE: We have added this content to the manuscript (lines 673-675).

  • COMMENT #4: Conduct a fibrosis regression analysis to account for confounding variables such as age, gender, race and family medical history.

     RESPONSE: This does not fall within the scope of this current review.

  • COMMENT #5: The most common factors increase the risks of fractures, slipped capital femoral epiphysis, and joint pain. Providing more details on these risks would benefit the reader.

     RESPONSE: We have added more detail with references as suggested (lines 301-309). 

 thank the reviewer and believe that the manuscript is improved as a result of their input.  We hope you will agree and decide in favor of accepting our report at this time.

Reviewer 4 Report

Comments and Suggestions for Authors

I thank the authors of the manuscript "From childhood obesity to metabolic dysfunction associated with fatty liver disease (MAFLD) and hyperlipidemia through oxidative stress" for the time they have devoted to conducting their research project and for the time they have devoted to writing their manuscript. This review aims to provide a broad summary of the common factors relating to childhood obesity, oxidative stress, insulin-resistance, and MASLD in the pediatric population. The manuscript is very interesting and well-written. The manuscript deals with a topical issue of interest to the scientific community. 

Even in the editorial form, the manuscript contains very few errors.

I ask the authors to correct lines 95,343,348,852 (I ask the authors to give a space. Example: to19 = to 19). 

I ask the authors to correct lines 103,119,122,172,245,247,249,256,270,286,308,564 (I ask the authors to control a space. Is there double spacing?). 

I ask the authors to correct line 199 (I ask the authors to add a point).

I ask the authors to correct lines 381 "[97,110]."

Table 1. In the "Examples" column, some rows show centered text and others show left-aligned text. Authors are asked to align the table (either all centered text or all left-aligned text).

In the text, some entries in the bibliography have a space between the numbers (see line 351) and others do not (see line 357). Authors are asked to standardize the drafting procedures.

Line 363: HDL-Cholesterol

The "p" of p value is always written in lower case. I ask the authors to correct the "p" every time they have written it in capital letters (P).

Line 603,607: <0.001 (Add 0)

Line 631: 6-19 years.

Line 841,845,848 = 95th (th = apice, apix) and m2 (2 = apice, apix)

Author Response

We thank the reviewer for thoroughly scrutinizing our manuscript. As requested, we have revised the manuscript and addressed the specific comments of the reviewer. The revised sections are delineated in red in a marked copy of the manuscript text.

Below, we provide a point-by-point response to the reviewer’s comments.

Reviewer # 4 Comments and Responses

  • COMMENT #1: I ask the authors to correct lines 95,343,348,852 (I ask the authors to give a space. Example: to19 = to 19).

        RESPONSE: Thank you for pointing out these errors. They have been corrected.

  • COMMENT #2: I ask the authors to correct lines 103,119,122,172,245,247,249,256,270,286,308,564 (I ask the authors to control a space. Is there double spacing?).

        RESPONSE: Thank you for pointing out these errors. They have been corrected.

  • COMMENT #3: I ask the authors to correct line 199 (I ask the authors to add a point).

        RESPONSE: This has been corrected.

  • COMMENT #4: I ask the authors to correct lines 381 "[97,110].".

        RESPONSE: This has been corrected.

  • COMMENT #5: Table 1. In the "Examples" column, some rows show centered text and others show left-aligned text. Authors are asked to align the table (either all centered text or all left-aligned text).

        RESPONSE: We have fixed the alignment of the table. 

  • COMMENT #6: In the text, some entries in the bibliography have a space between the numbers (see line 351) and others do not (see line 357). Authors are asked to standardize the drafting procedures.

                  RESPONSE: We have fixed these issues

  • COMMENT #7: Line 363: HDL-Cholesterol.

                  RESPONSE: We have changed this as indicated.

  • COMMENT #8: The "p" of p value is always written in lower case. I ask the authors to correct the "p" every time they have written it in capital letters (P).

        RESPONSE: We have made all “p” for p value lower case.

  • COMMENT #9: Line 603,607: <0.001 (Add 0).

                    RESPONSE: 0 added.

  • COMMENT #10: Line 631: 6-19 years.

                    RESPONSE: Corrected

  • COMMENT #11: Line 841,845,848 = 95th (th = apice, apix) and m2(2 = apice, apix)

                    RESPONSE: Corrected

 We thank the reviewer and believe that the manuscript is improved as a result of their input.  We hope you will agree and decide in favor of accepting our report at this time.